# Cobalt-Mediated Radical Copolymerization of Vinylidene Fluoride and 2,3,3,3-Trifluoroprop-1-ene

**DOI:** 10.3390/polym13162676

**Published:** 2021-08-11

**Authors:** Panagiotis G. Falireas, Bruno Ameduri

**Affiliations:** ICGM, Univ. Montpellier, CNRS, ENSCM, 34095 Montpellier, France; pfalireas@uliege.be

**Keywords:** 3,3,3-tetrafluoroprop-1-ene, cobalt acetylacetonate, controlled radical polymerization, kinetics of copolymerization, organometallic radical copolymerization, RDRP, reactivity ratios, vinylidene fluoride

## Abstract

New copolymers based on vinylidene fluoride (VDF) and 2,3,3,3-tetrafluoroprop-1-ene (1234yf) were synthesized by organometallic-mediated radical copolymerization (OMRcP) using the combination of bis(*tert*-butylcyclohexyl) peroxydicarbonate initiator and bis(acetylacetonato)cobalt(II), (Co(acac)_2_) as a controlling agent. Kinetics studies of the copolymerization of the fluoroalkenes copolymers were monitored by GPC and ^19^F NMR with molar masses up to 12,200 g/mol and dispersities (*Đ*) ranging from 1.33 to 1.47. Such an OMRcP behaves as a controlled copolymerization, evidenced by the molar mass of the resulting copolymer-monomer conversion linear relationship. The reactivity ratios, r_i_, of both comonomers were determined by using the Fineman-Ross and Kelen-Tüdos fitting model leading to r_VDF_ = 0.384 ± 0.013 and r_1234yf_ = 2.147 ± 0.129 at 60 °C, showing that a lower reactivity of VDF integrated in the copolymer to a greater extent leads to the production of gradient or pseudo-diblock copolymers. In addition, the Q (0.03) and e (0.06 and 0.94) parameters were assessed, as well as the dyad statistic distributions and mean square sequence lengths of PVDF and P1234yf.

## 1. Introduction

Fluoropolymers are outstanding specialty polymers which exhibit remarkable properties for High Tech applications [1,2,3,4]. Most fluorinated homopolymers display a high crystallinity rate (higher than 95% for PTFE) that makes them insoluble in many common organic solvents and further induces energetic processing costs to melt or sinter them [5]. Thus, copolymerizing fluoroalkenes overcome these limitations while also enabling better solution characterization. In fact, copolymers of vinylidene fluoride (VDF) have led to many studies and patents [6,7,8]. Among all VDF comonomers, 2,3,3,3-trifluoropropene, HFO-1234yf (CF_3_CF=CH_2_) is a valuable example of HFO class because it has a low global warming potential (GWP_100yrs_ < 1), a Montreal Protocol-regulated substance with high CO_2_-eq emissions, used in mobile air conditioning [9,10,11]. Indeed, a few radical copolymerizations of VDF with 1234yf have been reported, either by conventional radical copolymerization [12,13,14] or under reversible deactivation radical copolymerizations (RDRP), either by iodine transfer copolymerization [15] or RAFT [16]. However, to the best of our knowledge, although VDF has been successfully polymerized by cobalt-mediated radical polymerization (CMRP) [17,18], and although that technique can be well-applied on the copolymerization of vinyl acetate (VAc) [19], no study has been achieved on the CMR copolymerization of VDF with 1234yf. Wang et al. [20] reported the CMR copolymerization of chlorotrifluoroethylene (CTFE) with VAc initiated by AIBN in the presence of bis(acetylacetonato)cobalt(II), Co(acac)_2_, as a controlling agent. Though both molar masses and molar compositions of the resulting copolymers could be “controlled”, *Đ* values of the resulting copolymers were high (~1.57). Hence, the objective of this present study aims at copolymerizing VDF with 1234yf in controlled conditions by simply using a peroxycarbonate initiator and Co(acac)_2_ as a mediated compound to favor the control of such a reaction. Recently, the current authors have investigated and optimized the polymerization conditions of OMRP of VDF by redox initiation, initiated by bis(*tert*-butylcyclohexyl) peroxydicarbonate (P16) and Co(acac)_2_ as a controlling agent [17]. All polymerizations were carried out in dimethyl carbonate (DMC), a suitable solvent for PVDF oligomers that also exhibits fast rate of VDF polymerization [21]. By applying the same conditions, we investigated the copolymerization of VDF with 1234yf (Scheme 1). The results indicate that the process has the features of a controlled radical polymerization.

## 2. Results and Discussion

### Copolymerization of VDF with 1234yf

Recently, we reported that VDF could be homopolymerized by OMRP by redox initiation using the combination of P16 and Co(acac)_2_. The best control of the polymerization was found under optimized conditions ([P16]_0_/[Co(acac)_2_]_0_ = 2/1, T = 60 °C), leading to PVDF homopolymers with low dispersities (Ð < 1.35) [17]. Based on similar conditions, the copolymerization of VDF with 1234yf was performed, as illustrated by Scheme 1.

In order to get a deeper insight into the copolymerization of VDF with 1234yf, a kinetic study was performed covering the whole conversion of polymerization. Therefore, single-point experiments (ranging from 0.25 h to 16 h) were carried out. All copolymerizations were conducted under similar conditions ([VDF]_0_/[1234yf]_0_/[P16]_0_/[Co(acac)_2_]_0_ = 80/20/2/1 at 60 °C in DMC) and were monitored by ^19^F NMR (Figure 1) and GPC measurements equipped with a refractive index detector (Figure 2a). At this point, it should be stressed that in order to prevent unfavorable chain terminations, all polymerizations were quenched by TEMPO. According to the literature, TEMPO plays the role of a radical scavenger by displacing the covalently bonded Co(acac)_2_ moiety from the propagating active chains [22].

The GPC chromatograms of the synthesized copolymers exhibited negative refractive index (n_D_) signals since the n_D_ increment of PVDF (and probably poly(1234yf)) in DMF is negative [23,24,25]. Their profile displayed unimodal distributions showing no sign of coupling reactions and they significantly shifted toward lower elution times throughout the polymerization. Figure 2c illustrates the evolution of *M_n_* and Ð versus monomer conversion where the linear relationship between molar mass and conversion evidences that the chain growth has a controlled behavior. At the same time, dispersities increased progressively from 1.33 to 1.47.

The semilogarithmic kinetic plot exhibited two different regimes (Figure 2b). Initially, a fast polymerization was observed without any sign of induction period which is expected for a redox initiating system [26]. The copolymerization proceeded at a fast rate for 1 h, then it slowly decreased until it terminated after 8 h where the conversion reached 33%. Similar polymerization kinetics were found for homopolymerization of VDF by OMRP performed under the same conditions, where the change of polymerization rate was retarded after 2 h probably due to limited solubility of PVDF in DMC at 60 °C [17,18], as well as in RAFT polymerization of VDF [27].

By means of ^19^F NMR spectroscopy and based on previously reported studies, it was possible to thoroughly characterize the P(VDF-*co*-1234yf) copolymers synthesized by OMRcP after quenching with TEMPO and purification. 

The ^19^F NMR spectrum of P(VDF-*co*-1234yf) copolymers terminated at 8 h (Figure 3, Table 1, Entry 8) clearly displays a number of characteristic signals attributed to both comonomers: (i) the characteristic peaks ranging from −90.6 to −95.4 ppm correspond to the normal or Head-to-Tail (H-T) VDF-VDF dyads (-CH_2_-CF_2_-CH_2_-CF_2_-) in the PVDF chains, while those at −114.6 and −116.9 ppm are attributed to the reverse or Head-to-Head (H-H) VDF-VDF dyads (-CH_2_-CF_2_-CF_2_-CH_2_-) [6,7,23,25]; (ii) signals from −95.4 to −92.7 ppm are assigned to CH_2_-CF_2_-CH_2_-CF(CF_3_)-; (iii) the trifluoromethyl and tertiary fluorine groups of 1234yf in the -CH_2_-CF_2_-CH_2_-CF(CF_3_)- dyads of the copolymer are assigned from −78 to −82 ppm and in the −164 to −169 ppm range, respectively; (iv) and finally, closer inspection in regime between from −66 to −76 ppm revealed the presence of a series of peaks which could be attributed to fluorine groups of VDF or 1234yf adjacent to TEMPO end groups [17,18].

Additionally, the monomer reactivity ratios of the copolymers prepared by OMRcP were determined via the correlation of the copolymer-monomer feed composition relationship. Therefore, a series of seven P(VDF-*co*-1234yf) copolymers was carried out by OMRcP from initial [VDF]_0_/[1234yf]_0_ molar ratios ranging from 90/10 to 16/84 in DMC at 60 °C using a predetermined amount of P16 and Co(acac_2_) of which the molar ratio was always maintained at 2/1. The copolymer conversion was limited to less than 10–15% to minimize the drift of copolymer formation. The VDF molar fraction contained in the copolymer was assessed from ^19^F-NMR spectra based on Equation (1) (Figure 4, Table 2). 

It is obvious from the above analysis that the mole fraction of VDF in the copolymers is lower than in the feed (f_VDF_ > F_VDF_). This result clearly indicates that there is a significant difference in the reactivity ratios between the fluoromonomers. The reactivity ratios of VDF and 1234yf were determined by applying the Fineman-Ross [28] (Figure 5a) and Kelen-Tüdös [29] (Figure 5b) approaches. It is clear that regardless of the method chosen to estimate such parameters, the obtained data is rather similar in each case (Table 3). Specifically, the reactivity ratios are equal to r_vdf_ = 0.384 ± 0.013, r_1234yf_ = 2.147 ± 0.129 at 60 °C. As can be concluded from the calculations, the reactivity ratio of 1234yf is, in every case, significantly higher than that of VDF. The latter signifies that VDF propagation is not preferred in contrast to VDF and 1234yf cross-propagation while on the other hand, 1234yf prefers to undergo a homopolymerization rather than a copolymerization. Additionally, the product of the reactivity ratios for the copolymerization of VDF with 1234yf (r_vdf_×r_1234yf_) = 0.8) is close to 1, indicating a slight deviation from random polymerization kinetics and the formation of gradient copolymers of P(VDF-*co*-1234yf) copolymers. These results are in agreement with those of the literature in previous studies on P(VDF-*co*-1234yf) copolymers synthesized by iodine transfer polymerization and free radical polymerization where VDF is frequently reported as a less reactive monomer in copolymerization procedures [13,15], except with hexafluoropropylene [30] and perfluoromethylvinyl ether [31].

The parameters of specific reactivity (Q) and polarity (e) of a monomer refer to its stabilization by resonance and polar effects, respectively. To the best of our knowledge, as never reported in the literature, such parameters can be calculated for 1234yf using Alfrey and Price equations (Equations (1) and (2)) [32].
(1)$$r_{\mathrm{VDF}}=\frac{Q_{\mathrm{VDF}}}{Q_{1234\mathrm{yf}}}\exp[-e_{\mathrm{VDF}}\left(e_{\mathrm{VDF}}-e_{1234\mathrm{yf}}\right]$$
(2)$$r_{1234\mathrm{yf}}=\frac{Q_{1234\mathrm{yf}}}{Q_{\mathrm{VDF}}}\exp{\left[-e_{1234\mathrm{yf}}\left(e_{1234\mathrm{yf}}-e_{\mathrm{VDF}}\right)\right]}_{\;}$$

According to the literature, Q-e values of VDF have been reported to be 0.015 and 0.5, respectively [33]. These values, incorporated in Equations (1) and (2), enable the calculation of the respective Q-e values of 1234yf as follows: Q_1234yf_ = 0.04 and e_1234yf_ = 0.94 and Q_1234yf_ = 0.03 and e_1234yf_ = 0.06. As expected, a positive e_1234yf_ confirms that this monomer is electron-withdrawing (or acceptor), characteristic of fluoroalkenes, while a Q_1234yf_ value much higher than Q_VDF_ highlights a higher reactivity than VDF (or a lower activity of VDF growing radical).

The reactivity of each monomer can be illustrated by the statistical distribution of the dyad monomer sequences as VDF-VDF, 1234yf-1234yf, and VDF-1234yf, calculated according to the Igarashi method [34] based on Equations (3)–(5) (where φ_VDF_ represents the VDF mole fraction in the copolymer).
(3)$$f_{\mathrm{VDF}-\mathrm{VDF}}=\varphi_{\mathrm{VDF}}-\frac{2\varphi_{\mathrm{VDF}}\left(1-\varphi_{\mathrm{VDF}}\right)}{1+\sqrt{\left[{\left(2\varphi_{\mathrm{VDF}}-1\right)}^{2}+4r_{\mathrm{VDF}}r_{1234\mathrm{yf}}\varphi 1234\mathrm{yf}\left(1-\varphi_{\mathrm{VDF}}\right)\right]\;}}$$
(4)$$f_{1234\mathrm{yf}-1234\mathrm{yf}\;}=1-\varphi_{\mathrm{VDF}}-\frac{2\varphi_{\mathrm{VDF}}\left(1-\varphi_{\mathrm{VDF}}\right)}{1+\sqrt{\left[{\left(2\varphi_{\mathrm{VDF}}-1\right)}^{2}+4r_{\mathrm{VDF}}r_{1234\mathrm{yf}}\varphi 1234\mathrm{yf}\left(1-\varphi_{\mathrm{VDF}}\right)\right]\;}}$$
(5)$$f_{\mathrm{VDF}-1234\mathrm{yf}\;}=1-\varphi_{\mathrm{VDF}}-\frac{4\varphi_{\mathrm{VDF}}\left(1-\varphi_{\mathrm{VDF}}\right)}{1+\sqrt{\left[{\left(2\varphi_{\mathrm{VDF}}-1\right)}^{2}+4r_{\mathrm{VDF}}r_{1234\mathrm{yf}}\varphi 1234\mathrm{yf}\left(1-\varphi_{\mathrm{VDF}}\right)\right]\;}}$$

Mean sequence lengths PVDF and P1234yf were also calculated using the Equations (6) and (7) [35].
(6)μVDF=1+rVDF(MVDFM1234yf)
(7)μ1234yf=1+r1234yf(M1234yfMVDF)

The data are summarized in Table 4, whereas Figure 6 displays the variation of the dyad fractions versus the 1234yf mole fraction in the copolymer.

Figure 6 displays the variations of the dyad fractions versus the 1234yf mole fraction in the P(VDF-*co*-1234yf) copolymer. The results show that the mole fraction of the VDF-VDF dyad is constantly decreasing as the 1234yf mole fraction increases, while the same trend was also observed for VDF-1234yf dyads. On the other hand, the mole fraction of 1234yf-1234yf dyads increased gradually with 1234yf mole fraction. The above trend is a clear evidence of a great difference of monomer reactivity which was also verified by the calculation of the reactivity ratios values for the corresponding monomers.

## 3. Experimental Section

### 3.1. Materials and Methods

1,1-Difluoroethylene (VDF) and 2,3,3,3-tetrafluoropropene (1234yf) were kindly supplied by Arkema (Pierre-Bénite, France). Bis(*tert*-butylcyclohexyl) peroxydicarbonate (Perkadox^®^ 16, P16, 90%,) was obtained from AkzoNobel, reagentPlus grade; dimethyl carbonate (DMC, >99%, Merk, Darmstadt, Germany), Cobalt(II) acetylacetonate (Co(acac)_2_, 97%), 2,2,6,6-tetramethylpiperidine 1-oxy (TEMPO, 98%), and *n*-pentane (95%) were purchased from Sigma Aldrich and used as received. Deuterated acetone (acetone-*d_6_*) (purity > 99.8%) used for ^1^H and ^19^F NMR spectroscopy was purchased from Euroiso-top (Grenoble, France).

### 3.2. Characterizations

#### 3.2.1. Nuclear Magnetic Resonance (NMR) Spectroscopy

^19^F NMR spectra were recorded on a Bruker AC 400 Spectrometer (376 MHz for ^19^F) using acetone-*d_6_* as solvent. The sample temperature was set to 298 K. Chemical shifts and coupling constants are given in Hertz (Hz) and parts per million (ppm), respectively. The experimental conditions for recording the ^19^F NMR spectra were as follows: flip angle 30°, acquisition time 0.7 s, pulse delay 5 s, number of scans 64, and a pulse width of 5 μs for ^19^F NMR.

#### 3.2.2. Gel Permeation Chromatography (GPC)

The apparent numbers of average molar masses and dispersities of the synthesized polymers were determined using a GPC system (Varian 390-LC) multi-detector equipped with a differential refractive index detector (RI), using a guard column (Varian Polymer Laboratories, Church Stratton, UK, PLGel 5 μm, 50 × 7.5 mm), and two ResiPore columns of the same type. The mobile phase was DMF with 0.1 wt% LiBr adjusted at a flow rate of 1 mL min^−1^ while the columns were thermostated at 70 °C. The GPC system was calibrated using narrow poly(methyl methacrylate) (PMMA) standards ranging from 550 to 1,568,000 g/mol (EasiVial-Agilent, Stockpor, Cheshire, UK).

#### 3.2.3. OMRP of VDF with 1234yf Initiated by P16 in the Presence of Co(acac)_2_

The copolymerization of VDF with 1234yf was performed in a 50 mL Hastelloy autoclave Parr system (HC 276) equipped with a manometer, a mechanical Hastelloy anchor, a rupture disk (3000 PSI), inlet and outlet valves equipped with a special steel pipe, and a Parr electronic controller (for stirring speed and heating control). Prior to the introduction of the mixture solution, the autoclave was checked for any leaks by performing three vacuum-nitrogen cycles before finally applying vacuum (40 × 10^−6^ bar) for 30 min to remove any residual traces of oxygen. A typical copolymerization of VDF with 1234yf by OMRP mediated by Co(acac)_2_ was performed as follows (Table 1, Entry 8): initially, Co(acac)_2_ (0.40 g, 1.55 mmol) was introduced into the autoclave and then the reactor was closed and put under vacuum (10^−2^ mbar) so as to remove any residual traces of oxygen. Then, a degassed solution of DMC (30 mL) was transferred through a funnel tightly connected to the inlet valve of the autoclave. The reactor was then cooled in a liquid nitrogen bath, and subsequently VDF (8.34 g, 0.130 mol) and 1234yf (1.65 g, 0.014 mol) gases were introduced in sequence under weight control. Subsequently, the autoclave was progressively warmed up to 60 °C while the reaction solution was mechanically stirred. The polymerization was triggered by the introduction of a degassed solution of P16 (1.24 g, 3.11 mmol) in DMC (10 mL) in the reactor using an HPLC pump (5.0 mL/min). The copolymerization was conducted for 24 h and then was quenched by transferring (via a HPLC pump) a nitrogen-purged solution of TEMPO (0.480 g, 3.11 mmol, 2 equivalents with respect to Co(acac)_2_) in DMC (5 mL) into the autoclave and letting it react for 30 min at 64 °C, according to a previously reported procedure [1]. Finally, the autoclave was immersed in an ice bath, depressurized by venting, and opened to air. The purified copolymer was obtained after two repeated precipitations in 10-fold excess (400 mL) of chilled pentane and it was recovered by filtration followed by drying under vacuum overnight. The final product was recovered as a white powder (3.70 g, 34% yield) (in the case of higher 1234yf contents, whitish gums were produced) and characterized by ^19^F NMR spectroscopy and gel permeation chromatography. At this point, it should be mentioned that the polymerization yield was assumed identical as the monomer conversion, since it is particularly difficult to experimentally determine the comonomers conversion.

^19^F NMR (376 MHz, acetone-d_6_, δ (ppm), Figure 3): from −78 to −82 (-CF̲_3_ of 1234yf in the copolymer), from −90.6 to −95.4 (-CH_2_CF̲_2_CH_2_CF̲_2_-, normal addition of VDF); −95.4 to −92.7 (-CH_2_CF_2_-CH_2_CF(CF_3_)-) −114 (-CH_2_CF̲_2_-CF_2_CH_2_-CH_2_, reverse addition of VDF); −116 (-CH_2_CF_2_-CF̲_2_CH_2_-CH_2_, reverse addition of VDF); −165 (tertiary fluorine -CF̲(CF_3_) of 1234yf).

#### 3.2.4. Determination of the Reactivity Ratios of VDF and 1234yf

In order to determine the reactivity ratios of VDF and 1234yf by OMRP, seven copolymerizations were performed at different feed monomer compositions, with the VDF feed composition (f _VDF_) ranging from 0.16 to 0.90 (Table 2). Initially, solutions containing appropriate amounts of Co(acac)_2_, P16, and DMC were transferred via a metal syringe into a borosilicate Carius tube (length, 130 mm; internal diameter,18 mm; thickness, 2.5 mm; total volume, 16 cm^3^). The Carius tubes were then cooled in a liquid nitrogen bath and predetermined amounts of VDF and 1234yf gaseous monomers were transferred into the frozen tubes using a custom-made manifold that enables accurate measurement of the quantity of the gas (using “*pressure drop versus mass of monomer*” calibration curves). In all polymerization reactions, the concentration of the gaseous monomers to solvent was kept constant to 0.42 g/mL in order to ensure similar polymerization conditions. Subsequently, the bottleneck of the tubes was flame-sealed while keeping the content frozen in a liquid N_2_ bath. The polymerizations were started by immersing the tubes in a preheated and shaking bath thermostated at 60 °C. All copolymerizations were stopped after 15 min by freezing the tubes into liquid nitrogen and then opened to air, in order to ensure that the overall conversion of monomers was lower than 10%. The total product mixture was recovered, dried under vacuum overnight and characterized by ^19^F NMR spectroscopy in acetode-d_6_ for the determination of the molar fraction of VDF in the copolymer (F_VDF_) molar copolymer compositions (Figure 4 displays the stack of the ^19^F NMR spectra of all experiments). 

#### 3.2.5. Determination of Reactivity Ratios with the Fineman-Ross and Kelen-Tüdös Model

The molar fractions of VDF base units in the copolymer were determined using Equation (1) by taking the ratios of the integrals of all signals of −CF_2_ in VDF monomer units with respect to those of the signals of −CF_3_ in 1234yf units in the ^19^F NMR spectra: (8)$$\mathrm{mol}\%\;\mathrm{VDF}\;\mathrm{in}\;\mathrm{copolymers}=\frac{(\int_{-91}^{-96}{\mathrm{CF}}_{2\;}+\int_{-113}^{-118}{\mathrm{CF}}_{2})/2}{(\int_{-91}^{-96}{\mathrm{CF}}_{2\;}+\int_{-113}^{-118}{\mathrm{CF}}_{2})/2+\int_{-82}^{-84}{\mathrm{CF}}_{3}/3}\times 100$$

The reactivity ratios were determined by applying mathematical models of copolymerization which correlate the relationship between the composition of the monomer feed (f_VDF_) and the composition of the copolymers (F_VDF_). In the present study, Fineman-Ross (FR) and Kelen-Tüdös (KT) fitting curve methods were applied for the accurate calculation of the reactivity ratios. The former is expressed by Equation (2).
(9)$$G=H\times r_{1}-r_{2\;}$$

Plotting $G=\frac{f_{1}\left(2F_{1}-1\right)}{\left(1-f_{1}\right)F_{1}}$ as ordinate versus $H=\frac{f_{1}{}^{2}\left(1-F_{1}\right)}{{\left(1-f_{1}\right)}^{2}F_{1}}$ as abscissa yields a straight line, the slope of which represents *r*_1_ value while the intercept is −*r*_2_ value. The Kelen-Tüdös method employs the following Equation (3):(10)$$\eta =\left[\frac{r1+r_{2}}{\alpha }\right]\mu -\frac{r_{2}}{\alpha }$$
involving η and μ parameters which are mathematical functions related to the mole ratios in the monomer feed and in the copolymer and an arbitrary constant *α*. Such parameters are defined as $\eta =\frac{G}{\left(a+H\right)}$ and $\mu =\frac{H}{\left(\alpha +H\right)}$ and $\alpha =(H_{min}H_{max}$)^0.5^, where *H_min_* and *H_max_* are the highest and lowest values of *H* from the Fineman-Ross method. Thus, a plot of η versus μ gives a straight line which can be extrapolated at μ = 0 and μ = 1, thereby yielding r2a and $r_{1}$ as the respective η intercepts. 

## 4. Conclusions

For the first time, the OMR copolymerization of VDF and 1234yf initiated by the presence of P16 using Co(acac)_2_ as a controlling agent was successfully conducted. Polymerization kinetic was found to lead to an acceptable control under optimized conditions ([P16]_0_/[Co(acac)_2_]_0_ = 2/1, T = 60 °C), leading to P(VDF-*co*-1234yf) copolymers with relatively low dispersities (Ð < 1.47) until conversion up to 33%. The molar masses of such copolymers, determined by GPC, increased linearly with the monomer conversions. The reactivity ratios of both comonomers were calculated based on Fineman-Ross and Kelen-Tüdös models where similar ratios were obtained in both cases: r_VDF_ = 0.384 ± 0.013 and r_1234yf_ = 2.147 ± 0.129 at 60 °C, denoting that 1234yf is more reactive than VDF, thus likely favoring the formation of gradient copolymers. The result is in agreement with the dyad statistical distribution in the copolymers. Finally, the Q-e values could also be calculated for the first time. The above results demonstrate that Co(acac)_2_-mediated controlled radical copolymerization of VDF and 1234yf initiated by P16 is possible. Finally, future works will look more closely into further optimization of this synthetic strategy which could open the route towards the synthesis of well-defined block terpolymers that will offer the desired combination of properties for advanced applications.

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
