# Peer review of "Cobalt-Mediated Radical Copolymerization of Vinylidene Fluoride and 2,3,3,3-Trifluoroprop-1-ene"

_polymers, 2021, doi:10.3390/polym13162676_

Round 1

Reviewer 1 Report

The synthesis of statistical copolymers of vinylidene fluoride and 2,3,3,3-trifluoroprop-1-ene by cobalt-mediated radical polymerization is reported in this manuscript. There is a sufficient degree of originality, since the specific polymerization technique has not been previously reported for the synthesis of these statistical copolymers. The conclusions are well supported by the experimental findings. The most important comments, suggestions and criticisms on this work are given below:

  • Table S1 and Figures S1 and S2 are incorporated in the main body of the manuscript, despite the fact that they belong to the Supporting Information Section.
  • The Fineman-Ross and Kelen-Tüdos methods are not the only ones for the determination of the monomer reactivity ratios. In fact, the non-linear regression methods are considered as more accurate to describe the copolymerization behavior. The authors are advised to employ none of these approaches.
  • Since the authors have measured the reactivity ratios it would be very easy for the authors to provide more structural data for the copolymers, such as the distribution of dyad sequences and the mean sequence monomer length.

Author Response

Reviewer 1  The synthesis of statistical copolymers of vinylidene fluoride and 2,3,3,3-trifluoroprop-1-ene by cobalt-mediated radical polymerization is reported in this manuscript. There is a sufficient degree of originality, since the specific polymerization technique has not been previously reported for the synthesis of these statistical copolymers. The conclusions are well supported by the experimental findings. The most important comments, suggestions and criticisms on this work are given below:

  • Table S1 and Figures S1 and S2 are incorporated in the main body of the manuscript, despite the fact that they belong to the Supporting Information Section.

Answer: we are sorry for these mistakes and have made the suitable corrections in the revised main manuscript

  • The Fineman-Ross and Kelen-Tüdos methods are not the only ones for the determination of the monomer reactivity ratios. In fact, the non-linear regression methods are considered as more accurate to describe the copolymerization behavior. The authors are advised to employ none of these approaches.

Answer: there are several methods to assess the reactivity ratios (including non-linear regression and linear approaches). We used the Mayo Lewis method, but that method led to strange results while we have not tried more recent methods (e.g. from Sir Jenkins, Macromolecules 1985, 18, 6, 1241–1244) which require to copolymerize these comonomers with polar and unpolar monomers. Indeed, this method will deserve a deeper work and unfortunately, we donot have 1234yf monomer anymore.

  • Since the authors have measured the reactivity ratios it would be very easy for the authors to provide more structural data for the copolymers, such as the distribution of dyad sequences and the mean sequence monomer length.

Answer: We thank the reviewer for this useful suggestion. The distribution of dyad sequences and mean sequence monomer length have been calculated and have been supplied in the revised manuscript.

Reviewer 2 Report

The manuscript writing by Panagiotis G. Falireas and Bruno Ameduri show an organometallic-mediated radical copolymerization (OMRcP) method to synthesize fluoroalkenes copolymers based on vinylidene fluoride (VDF) and 2,3,3,3-tetrafluoroprop-1-ene 7 (1234yf). The resultant copolymers give molar masses up to 12,200 g/mol and dispersities (Đ) ranging from 1.33 to 1.47. Kinetics studies demonstrate that the OMRcP behaves in a controlled manner. The reactivity ratios of both comonomers were calculated carefully. In brief, the manuscript is interesting and well written, and thus publication in Polymers can be recommended after address the following concerns from myside:

(1) The author should explain why the conversion is so low (33%) in the current OMRcP method.   

(2) As a controlled radical polymerization, it would be more interest to the chain extension/re-initiating in OMRcP as well as the synthesis of a block copolymer based on VDF and 1234yf.  

(3) At least, a 1H NMR spectrum of the copolymer should be offered to demonstrate the presence and content of TEMPO in the end chain.

Author Response

Reviewer 2 The manuscript writing by Panagiotis G. Falireas and Bruno Ameduri show an organometallic-mediated radical copolymerization (OMRcP) method to synthesize fluoroalkenes copolymers based on vinylidene fluoride (VDF) and 2,3,3,3-tetrafluoroprop-1-ene 7 (1234yf). The resultant copolymers give molar masses up to 12,200 g/mol and dispersities (Đ) ranging from 1.33 to 1.47. Kinetics studies demonstrate that the OMRcP behaves in a controlled manner. The reactivity ratios of both comonomers were calculated carefully. In brief, the manuscript is interesting and well written, and thus publication in Polymers can be recommended after address the following concerns from myside:

(1) The author should explain why the conversion is so low (33%) in the current OMRcP method.

Answer: As it is mentioned in the manuscript (page 7), we strongly believe that such a low conversions is due to the limited solubility of the increased molar mass of such highly fluorinated P(VDF-co-1234) copolymers in DMC. Similar results were found by our group for the polymerization of VDF by OMRP (Macromolecules 2019, 52 (3), 1266-1276 and Angewant. Intern. Ed. 2018, 57 (11), 2934-2937) as well as with RAFT Macromolecules 2016, 49 (15), 5386-5396

(2) As a controlled radical polymerization, it would be more interest to the chain extension/re-initiating in OMRcP as well as the synthesis of a block copolymer based on VDF and 1234yf. 

Answer: Yes, this is a good statement for an additional clue of controlled radical (co)polymerization but we have not attempted it, though we previously reported (reference 17) that CMRP enables to prepare block copolymers.

(3) At least, a 1H NMR spectrum of the copolymer should be offered to demonstrate the presence and content of TEMPO in the end chain.

Answer: The reviewer is right. Unfortunately, the quality of all 1H NMR spectra was not as good as expected. Therefore, we could not extract any useful information to highlight signals assigned to TEMPO.